# Physical Activity of Serbian Children in Daycare

**DOI:** 10.3390/children8020161

**Published:** 2021-02-21

**Authors:** Milenko Janković, Maja Batez, Dušan Stupar, Jelena Obradović, Nebojša Trajković

**Affiliations:** 1Preschool Teacher Training School, University of Novi Sad, 21000 Novi Sad, Serbia; milenkojankovic@live.com; 2Faculty of Sport and Physical Education, University of Novi Sad, 21000 Novi Sad, Serbia; jdobradovic@sbb.rs; 3Faculty of Sport and Tourism, University of Novi Sad, 21000 Novi Sad, Serbia; dusan.stupar@tims.edu.rs; 4Faculty of Sport and Physical Education, University of Niš, Serbia, 18000 Niš, Serbia; nele_trajce@yahoo.com

**Keywords:** physical activity, daycare, children

## Abstract

Background: Monitoring of physical activity within the educational institution is of great importance, primarily because of the orientation and content implemented in the daycare. This research aimed to examine the number of steps children took during their stay in daycare with regards to age, gender and the frequency of going out. Methods: The research was conducted in four daycares in the urban environment of Novi Sad (Republic of Serbia), where 231 children, aged 5 to 7, were monitored (129 boys and 102 girls). Data on the number of steps were obtained using the pedometers. Results: The result of the univariate analysis of the variance test confirmed a statistically significant difference in the number of steps in relation to the age of children (*p* = 0.04). Boys were more active than girls in both age groups (*p* = 0.001). Children who were going to the daycare yard three to five times a week took drastically more steps than children who went up to twice a week (*p* = 0.001). Conclusion: The results of the current study show that age, gender, and time spent outdoors are significant determinants of physical activity in preschool age. Therefore, interventions regarding physical activity should be made during early childhood in order to promote health and prevent disease.

## 1. Introduction

The growth and development of children from the seventh to the eighth year involves complex processes of biological and psychosocial changes. Daily moderate-to-vigorous physical activity is of great importance for proper childhood growth and development [1] and, together with proper nutrition, it may prevent obesity and accordingly help avoiding many health problems. A broader definition of physical activity involves any physical movement produced by skeletal muscles that leads to energy consumption [2]. It was stated earlier that physical activity of children aged 5 to 17 includes games, sports, transport forms of movement, recreational activities, and activities during physical education [3].

However, physical activity is a variable category because its level and intensity are not the same in children of different ages [4], boys are more physically active [5,6,7], children of physically active parents are more active [6], which also holds true for children of parents with a higher level of education [8]. It also depends on the season [9,10] time spent outdoors [4,11], environmental management [12]. On the other hand, sedentary behavior includes activities like sitting, reclining, or lying during which ≤1.5 metabolic energy is being spent [13].

The preschool age group was considered as extremely physically active earlier, but it was recently determined that their activities were predominantly sedentary [14]. Children aged 4 and 5 daily spend 85% of their time in sedentary activities and are engaged in moderate to intense physical activity only during 5% of the time [15]. The new recommendations of the World Health Organization [3] state that children aged 5 to 17 need to have at least 60 min of moderate to strong and intense physical activity daily. Moreover, new guidelines have included sedentary behavior guideline recommendations. If recommendations on PA are made on the basis of the number of steps taken daily, then boys aged 6 to 12 should take about 15,000 steps and girls should take about 12,000 [16]. Bearing in mind that physical activity is one of the factors that affects obesity, similar values have been suggested [17], stating that boys should take around 16,000 steps a day and girls should take around 13,000 in order to have a normal body fat percentage.

Active participation of educators in the games, more frequent involvement of parents in the physical activity of children, and an adequate behavioral model for the children to look up to are just some of the recommendations that contribute to a higher level of physical activity of children [18]. Bearing in mind the amount of time that children spend there, daycare is extremely important for exercising and promoting physical activity. Additionally, educators in daycare play a significant role, because a positive correlation between their personal engagement and children’s physical activity has been established [19]. For all the mentioned reasons, the time children spend in the institution should satisfy or partially meet the daily recommendations for their physical activity.

In studying the physical activity of children within the educational institution, it has been established that children spend most of their time in sedentary activities (80%), and only 3% of the time in moderate to intensive physical activities [20]. It was also determined that first-grade students achieve a greater number of steps during the day in the institution than children in daycare; that boys move more; that greater movement activity is achieved if organized physical activity is carried out during the day and if the break is spent outdoors [21]. A study which was carried out in Sweden concluded that the older children achieved higher number of steps; it was found that children spent an average of 7 h and 22 min in the preschool institution, and during that time, they took an average of 7313 steps [22].

Although the accelerometers are considered as a golden standard in measuring physical activity, pedometers are more likely to be adopted for public health applications. Moreover, they have easy interpretability and are relatively cheap. Accordingly, such good and large reference data could lead to effective public health guidelines. In Serbia, daycare attendance is mandatory for children between 4–6 years of age, with most of the children participating in pre-school education for 6–8 h/day. Therefore, level of physical activity during daycare time is of special interest. Moreover, monitoring the physical activity of children is very important because interventions can be made in daily and weekly content according to the obtained data. This study aimed to examine the activity of preschool children during their stay in the institution, and based on it, determine the results in relation to children’s age, gender, and the frequency of going out. It was hypothesized that children in daycare would show differences according to age, gender and time spent outdoors.

## 2. Materials and Methods

### 2.1. Participants

Data collection was carried out from 28 April to 30 May 2014, in four daycares located in the urban environment of Novi Sad (Republic of Serbia). The daycares where the survey was conducted had good conditions for indoor activities (rooms in which children stayed were spacious, they had a physical exercise room, etc.) and outdoor activities (a yard within the daycare). In this study, 231 children, or 129 boys and 102 girls, were monitored. The average age of the children was 6.1 years, ranging from 5.3 to 6.9 years. The parents gave a written approval for the participation of their children in the tests. This study was approved by the Institutional Review Committee of the University of Novi Sad (04-29/1) and Faculty of Sport and Physical Education (01-258/1) and it was conducted under the Declaration of Helsinki.

### 2.2. Procedures

Physical activity was monitored using the pedometer Yamax Digiwalker CW-700 (Yamax Corporation, Tokyo, Japan), whose reliability was checked in previous studies on children [21,22,23]. Data on physical activity were derived from the pedometer used in previous studies on children [24,25,26,27]. Pedometers are used to detect ambulatory activity (walking) and are insensitive to nonlocomotor forms of movement. Moreover, pedometers are not able to discriminate for intensity. Nevertheless, the record of daily steps can be a significant marker to measure and track children. Pedometers were sealed and placed in the area of the belt to the left in a way that did not interfere with the children while exercising during their stay in the daycare.

Before the start of the research, the management of the preschool institution was informed about the research protocol, after which their consent for implementation of the research was obtained. Parents were informed of the protocol and procedures for tracking the number of steps using the pedometer, which resulted in their consent for their child to participate in the research. The number of children’s steps during their stay in the daycare was monitored during five consecutive working days, and the final result is the sum of all registered values. If the child was absent during all five working days, his/her result was not included in the analysis. The attaching of the pedometer was done at the beginning of the working day, when the child came to the daycare, and the result was read right before the child left daycare. In addition to recording the number of steps, the time when the child came and when the child left the daycare, as well as the minimum and maximum temperature of the day was also registered. The children wore a pedometer for five working days, for an average of 30 h and 38 min. Thus, the average daily wearing of a pedometer was about six hours (6 h 8 min).

Pedometers were used from the moment the children arrived in daycare until they left the daycare. All daycares in which the research was conducted had adequate conditions for the implementation of various contents of physical activities and safe fenced yard. In addition, during the period of children’s stay in the kindergarten, there was a possibility to organize a walk outside the facility. The protocol was not followed in a negligible number of cases due to the removal of the pedometer by the children. However, due to good organization and constant monitoring of assistants, there were no complications, and in cases of removal of pedometers, the results were not taken into account when processing the data. Each child had their own pedometer to wear during the day, so there was no need to rotate the pedometers.

During the children’s stay in, daycare, they had free time and learning activities in various fields (art, music, math, science, social studies, literature). In addition to the above, one part of the day was spent on health and hygiene activities such as: maintaining personal hygiene, nutrition, sleeping, cleaning the room, etc. The arrival of children in daycare was possible from 6 o’clock in the morning, and they could stay until 5 o’clock in the afternoon, so the children spent on average of 7 h in the daycare daily. Such a possibility (arrival from 6 o’clock) exists because of the parents who have to start their working activities earlier, but the largest number of them (>90%) come in the period from 7:30 to 8:00 o’clock. The daycares where the research was conducted had a complete infrastructure adapted to children (study room, toilet, dining room, gym, fenced yard, etc.), so that the children could move around the daycare unhindered and safely.

An example of a typical day in the daycares where the research was conducted included: the arrival of children (from 6:00 to 8:00 h), breakfast (from 8 to 8:30 h), free activities or indoor activity (including teacher-directed games) (from 8:30 to 10:00 h), snack (10:00 to 10:15 h), learning activities (example of activities: creative art, cooking, science/discovery, dramatic play, language art/listening, sand and water play, dramatic role play, fine motor manipulatives, gross motor skills) (10:15 to 11:00 h), rest-children are required to rest for a reasonable period but not required to sleep.(11:00 to 12:00 h), lunch (12:00 to 13:30 h), leisure time-outdoor play or indoor play, free choice in activity areas, preparations for departure (14:00 to 17:00 h).

### 2.3. Statistical Analysis

The data collected were processed in the Statistical Package for the Social Sciences version 20.0. A Kolmogorov–Smirnov test was used to examine whether the variables were normally distributed. All the data were normally distributed, so parametric statistics was used for further analysis. The values of the arithmetic mean and standard deviation are presented for the total number of steps taken after five day period. The difference in the number of steps between groups in relation to age, gender affiliation, and frequency of going out was established by univariate variance analysis (ANOVA). Cohen d effect sizes (ES) were calculated also to determine the magnitude of the group differences in number of steps. The ES were classified as follows: <0.2 as trivial, 0.2–0.6 as small, 0.6–1.2 as moderate, 1.2–2.0 as large, and >2.0 as very large.

## 3. Results

Table 1 shows the value of the number of children’s steps in relation to their age, and the combined values of boys and girls.

The result of the univariate analysis of variance test confirms a statistically significant difference in the number of steps in relation to the age of children (*p* = 0.04) with small ES (0.268). Based on the results of both group’s means, it can be concluded that the younger children took more steps than the older ones. The difference in the number of steps on a weekly basis is around 1250, which is about 250 steps on a daily basis.

Further analysis established the difference in the number of steps taken between boys and girls in both age groups (Table 2).

There is a statistically significant difference between boys and girls in the number of steps taken during their daycare stay in children aged 5 years (F_1.229_ = 46.21; *p* = 0.001) with moderate ES (0.896) as well in children aged 6 years (F_1.229_ = 26.94; *p* = 0.001) with moderate ES (0.684). The average values of the number of steps indicate that boys aged 5 years take about 4500 steps more than girls per week, and children aged 6 years take about 2500 more. So, the difference is present in both age groups, and amounts to between 500 and 900 steps on a daily basis.

Educational activities in daycares are usually carried out in study rooms, but educators have always used fine weather to take the children to the daycare yard or to a nearby park. Table 3 shows a number of steps taken depending on how often during the week the children were taken outdoors.

Regular stay of children outdoors contributed to their greater movement, which is confirmed by Anova that showed significant differences (F_1.229_ = 46.21; *p* = 0.001) in number of steps concerning the number of days spent outdoors. The average values in Table 3 indicate that a higher level of movement was achieved if children went out between three to five times a week (from 25,802 to 28,340 steps during the week).

## 4. Discussion

The aim of the current study was to examine the number of steps children took during their stay in daycare with regards to age, gender, and the frequency of going out. The outcome is that that age, gender, and time spent outdoors are significant determinants of physical activity in preschool age. During their stay in the daycare, children participated in activities from different educational areas according to the educator’s curriculum and in free activities. All activities were conducted in the room, exercise room or outdoors (daycare yard, park, street, etc.). The policy of the preschool institution and the implemented program present a potential that could significantly affect the overall physical activity [5,28]. Therefore, it is necessary to insist on adapting the curriculum of the daycare to the current lifestyle. Moreover, having an adequate room in daycare is of great importance for those who may not have opportunities to participate in organized sports or structured physical education classes. Accordingly, the curriculum should be adapted according to conditions in daycares. The conditions and curriculum in the current study were very similar in all daycares, which allow us to make conclusions regardless of different factors that could impact the physical activity during daycare.

In previous studies, contradictory data on the activity of children, depending on age, were found. Some studies found that older children took more steps during their stay in an educational institution [21,22], and some found that these results were higher in younger children [20]. This study was carried out in daycares that are about the same size and with similar equipment (containing physical exercise room, instruments and equipment, a secured courtyard, etc.). Children of all age groups spent time in such conditions, so it could be stated that they were all in almost the same conditions. It was assumed that children of younger age needed to take several steps to perform certain task because they were smaller, their step was shorter and less rational than the one of older children. So, when moving around the study room, daycare, or yard, the older children needed fewer steps to cover the same distance as the younger children. This may be one of the reasons why younger children achieved higher number of steps, but additional research should be carried out to arrive at a reliable conclusion.

Children’s steps represent one segment of physical activity and could be an integral part of free or organized physical activity and children’s locomotion in performing various tasks. Although boys and girls participated in activities together and stayed in the same area, there was a difference in the number of steps. The results of previous research clearly confirmed that boys took more steps throughout the day [24,25,26,29,30]: this information was also obtained in the research in which children’s movements in preschool institutions were monitored [5,21,22], which was also confirmed by the results of this research.

It is known that boys spend more time playing outdoors [31] and more often participate in sports activities conducted in a larger group, unlike girls who prefer locomotor activities in smaller groups [32]. The assumption was that the biggest difference between boys and girls in activity level was made during free activities, as boys exercised dynamic games and games in which there is more locomotion more often. All of the above, as well as the different interests of boys and girls, contributed to the fact that boys took more steps during their stay in daycare. The difference in the level of activity between boys and girls was partly based on biological factors since it was also observed in newborns [33]. As time passes, the gap widens due to the interaction with environmental factors in which children grow and developed [34], but also due to children’s socialization and different expectations regarding boys and girls [33]. The further process of their biological development should also be taken into consideration.

The environmental factor is very important when it comes to frequency and time spent in physical activities. In previous studies, as well as in this one, it was found that staying outside significantly contributes to the greater ranges of locomotion of children [12,35], since the environment in which children move about can motivate them to be physically active. The importance of the environment can already be seen with infants, at 12 months of age, where a negative correlation between the limited space for locomotion in daycare and the level of physical activity and body composition exists [36]. The possibility of adapting the child’s environment to stimulate the exercise of physical activity was confirmed by studies that found that the child’s physical activity was significantly higher if there was more portable play equipment in the daycare, larger playgrounds and less electronic media [37,38]. This study confirms the earlier findings that the amount of space and outdoor activities play a significant role in children’s physical activity during their stay in daycare.

The level and intensity of physical activity could be increased through different interventions, even in indoor settings [39]. It is necessary to point out the results of this and similar research, that the frequency of going out significantly increases the level of motor competences and locomotion [40]. It has also been confirmed that providing the ability to use portable play equipment and short training of educators can contribute to a higher level of locomotion of children in daycare [41]. The amount of physical activity was different by sex, showing that girls were less physically active during their stay in daycares. A possible explanation for the differences could be that girls were more engaged in gender-specific activities. Therefore, children in daycares should engage in neutral activities in order to prevent these differences from early childhood. Moreover, attention should be paid to girls and their physical activity should be stimulated, as it was found that they were less active throughout the day and during their stay in the institution. Another explanation for gender differences could be the fact that girls showed lower motor competence than boys in this age group [40], and motor competence is strongly associated with physical activity [41].

The priority of the study is emphasized by one segment of physical activity (number of steps) during the stay of children in the daycare, where they spent about six hours daily. In addition, the results being shown in relation to age, gender, and the frequency of going out represents an advantage. The results of this study were collected using a pedometer that is relevant and often used as a measuring instrument in this type of research. The fact that no parents’ behavior or socioeconomic factors were included should be taken as a limitation of the current study. Moreover, even though children had similar conditions, they were not compared in the current study, which could be another limitation. Therefore, future studies should focus on daycare conditions and possibilities to increase physical activity having in mind the amount of time they spend in daycare. The major limitation is the fact that pedometers cannot provide information about the intensity of physical activities. However, having in mind that children spend most of the time in daycare and daycare yard, activities with higher intensities were not so frequent. Additionally, sample of 231 children provide a sufficient picture of these children’s activity during daycare. However, drawing the sample from daycares in Northern Serbia only does not allow generalization of the current results to the Serbian population. Nevertheless, taking account the interests in daily physical activities in preschool children and the fact that this is, according to authors knowledge, the first study of such in Serbia, represents the biggest strength of this study.

## 5. Conclusions

The current results confirm the hypothesis that differences exist in the number of steps taken regarding the age, gender and time spent outdoors. Moreover, this research showed that the number of children’s steps during daycare stay were far from the recommended standards by referential world organizations. Nevertheless, it was established that regular outdoors and structured physical activity contribute to the greater range of locomotion of children. Children spend a significant amount of time throughout the day in daycare, so it is important to determine how physically active they are during their stay and what influences the level, intensity, frequency and content of a physical activity. The recommended value of the level and intensity of physical activity for children should be achieved or approximately achieved during their stay in the daycare. This would influence most of preschool-age children and there would be a possibility for the educators and the teachers of physical education to control and monitor physical activities.

## Figures and Tables

**Table 1 children-08-00161-t001:** Number of children’s steps according to age. Values are means and standard deviation (SD).

	Children Aged 5 Years (*n* = 122)	Children Aged 6 Years (*n* = 109)	F_(1.229)_	*p*	Effect Size (Cohen’s d)	95% C.I.
Number of steps	23,861.9 ± 5130.5	22,612.5 ± 4116.6	4.10	0.04	0.268	0.007–0.526

**Table 2 children-08-00161-t002:** Number of steps according to age and gender. Values are means and SD.

	Children Aged 5 Years		F_(1.120)_	*p*	Effect Size (Cohen’s d)	95% C.I.
	Boys	Girls				
Number of steps	25,617.2 ± 5530.7	21,196.3 ± 4167.1	46.21	0.001	0.896	0.625–1.1669
	**Children Aged 6 Years**					
	**Boys**	**Girls**				
Number of steps	23,888.4 ± 3970.4	21,213.9 ± 3842.2	26.94	0.001	0.684	0.2976–1.0709

**Table 3 children-08-00161-t003:** Number of steps according to the frequency of going outdoors.

Number of Days Outdoors		Mean ± SD	F	*p*
0 ^a^	(*n* = 27)	19,558.3 ± 2736.6	37.17	0.001
1 ^b^	(*n* = 37)	20,168.6 ± 2224.3		
2 ^ac^	(*n* = 61)	21,155.9 ± 3520.3		
3 ^abcd^	(*n* = 56)	25,802.4 ± 2979.6		
4 ^abcd^	(*n* = 14)	28,340.4 ± 3984.5		
5 ^abc^	(*n* = 36)	26,622.3 ± 4658.9		

^a^ statistical significance between 0 and 2, 3, 4, 5; ^b^ statistical significance between 1 and 3, 4, 5; ^c^ statistical significance between 2 and 3, 4, 5; ^d^ statistical significance between 3 and 4.

## Data Availability

The data sets generated during and/or analyzed during the current study are available from the corresponding author on reasonable request.

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
