# Peer review of "Physical Activity of Serbian Children in Daycare"

_children, 2021, doi:10.3390/children8020161_

Round 1

Reviewer 1 Report

Comments for Manuscript ID children-1094496:

  1. The first question for the authors is what was the novelty of the study? the authors must argue in the Introduction chapter this aspect.
  2. Lines 45, 47, 48, 57, 64 - too much space between words, please correct.Also, why the article is not put in new template?
  3. ''Data collection was carried out from April 28 to May 30, 2014'', Isn't this research too old? or the period? it's been 7 years since then, I think something current would have been more interesting.
  4. Where the pedometers used only during the stay at the kindergarten or outside it? Did the children have the patience to use them, didn't they try to take them down during the activities? Did each of the children have their own pedometer, or were they used by rotation? the authors must detailed
  5. ''The arrival of children in daycare was possible from 6 114 o'clock in the morning, and they could stay until 5 o'clock in the afternoon'', 6 o'clock isn't it too early to come to kindergarten? why did the activity start at this time? please detailed
  6. I think the results chapter should be improved with other data, it is too simple present. A more detailed analysis is needed, please also perform an analysis of data by age and include a detailed table with this to see the dynamics of physical activity by age.
  7. I did not understand very clearly if the authors presented the limits of the paper, please highlight this aspect in the Discussions chapter.
  8. References - please put in accordance with the policy of journal, please see the template instructions.

Author Response

February 01, 2021

Re: Children- 1094496

To the Editors

Dear Editor,

Please find attached the revised version of our manuscript children- 1094496, entitled “Physical Activity of Serbian Children in Daycare”

We thank the reviewers for their careful evaluation and helpful comments to our manuscript. We have carefully taken their comments into consideration in preparing our revision, which has resulted in a paper that is clearer, broader and more compelling.

Some changes were made to this paper in order to improve it. Please find below our point-by-point responses to each of the comments of the reviewers. All changes are marked in the text.

We hope that the revisions in the manuscript and our accompanying responses will be sufficient to make our manuscript suitable for publication in Children.

Sincerely,

Maja Batez

Reviewer 1

Thank you for your detailed review of our manuscript and for providing some insightful and thought-provoking suggestions to strengthen our manuscript. We feel we have sufficient responses to each of your major concerns listed above, which are further detailed below, and hope that they alleviate the concerns you have regarding the approaches adopted in our manuscript.

  • The first question for the authors is what was the novelty of the study? the authors must argue in the Introduction chapter this aspect.

Our response: We are sorry for the missing part that is of great importance. We have added the importance of the study. Moreover, the novelty is that there is no such study or data in Serbia which was stated later in the discussion part. The change was marked yellow in the introduction and below also.

Although the accelerometers are considered as a golden standard in measuring physical activity, pedometers are more likely to be adopted for public health applications. Moreover, they have easy interpretability and are relatively cheap. Accordingly, such good and large reference data could lead to effective public health guidelines.  In Serbia, daycare attendance is mandatory for children between 4–6 years of age, with most of the children participating in pre-school education for 6–8 hours/day. Therefore, level of physical activity during daycare time is of special interest.

  • Lines 45, 47, 48, 57, 64 - too much space between words, please correct.Also, why the article is not put in new template?

Our response: Sorry, this was because of many track changes from earlier corrections. It was corrected.

  • ''Data collection was carried out from April 28 to May 30, 2014'', Isn't this research too old? or the period? it's been 7 years since then, I think something current would have been more interesting.

Our response: We agree with you. However, in Serbia the University rules and law doesn’t allow you to publish data before phd defense. The defense was later in 2016. Secondly, the paper was submitted in two top 5% journals. Unfortunately, it was rejected but the time spent in journals was more than a year. Nevertheless, according to authors knowledge, there are still no such studies in Serbia so this paper still has its value.

  • Where the pedometers used only during the stay at the kindergarten or outside it? Did the children have the patience to use them, didn't they try to take them down during the activities? Did each of the children have their own pedometer, or were they used by rotation? the authors must detailed

Our response: We have added the requested information regarding the protocol and possible removal of the pedometers. The changes were marked yellow in the methods part.

  • ''The arrival of children in daycare was possible from 6 114 o'clock in the morning, and they could stay until 5 o'clock in the afternoon'', 6 o'clock isn't it too early to come to kindergarten? why did the activity start at this time? please detailed

Our response: Bringing children to daycare was possible from 6 o'clock. Such a possibility exists because of the parents who have to start their working activities earlier. It was clearly stated that the arrival of children is from 6 to 8 o'clock, but the largest number of them come in the period from 7:30 to 8:00 o'clock. We have additionally explained this in the methods part and marked yellow.

  • I think the results chapter should be improved with other data, it is too simple present. A more detailed analysis is needed, please also perform an analysis of data by age and include a detailed table with this to see the dynamics of physical activity by age.

Our response: We totally agree with your comment. However, more data and analysis could not be added due to the fact that it is used in another manuscript that is under review. Moreover, this study was more descriptive in order to show the reference values and for possible comparison with similar studies. We could divide the groups to half year groups but the sample per group would be significantly smaller. Therefore, although the analysis seems simple, in this moment it represents a valuable information regarding physical activity in Serbia. If you have any specific recommendation for different analysis, please feel free to suggest and we will try to answer to the request.

  • I did not understand very clearly if the authors presented the limits of the paper, please highlight this aspect in the Discussions chapter.

Our response: We agree that limitations were not highlighted and clearly explained; therefore we added the important limitations and also the biggest strength of the study. The changes were marked yellow in the last paragraph of discussion.

  • References - please put in accordance with the policy of journal, please see the template instructions.

Our response: It was corrected, thank you.

Reviewer 2 Report

Thank you for making the changes.

Author Response

Thank you for your detailed review of our manuscript and positive comments. 

Round 2

Reviewer 1 Report

Unfortunately, the authors' answers to my comments were not convincing, the most important part of this article was not modified / supplemented with new data, I am referring to the results chapter. The authors present, as I have said some simple statistical analyzes, i.e. SD, p. and F, I think it is very little for an article and for such a reputable journal. The authors justified that they cannot add other data because they have other articles in evaluation, the motivation is not enough, I consider that the results of a research are essential and especially the degree of complexity for a scientific article. Taking into account these aspects, I consider that the article must be reviewed again and completed with what I requested.

The authors also said that they changed the presentation of the references, if we look at the references they do not agree with the policy of the journal, see the abbreviation of the journal and others.

Also, I did not understand the explanation regarding the fact that some children arrive at kindergarten at 6.00, please a new detail, I think it is very hard to believe to get to kindergarten at that time, especially for children.

Author Response

Dear Editors,

Thank you for providing us this opportunity to further revise our manuscript entitled Physical Activity of Serbian Children in Daycare.

We appreciate the very positive and constructive comments from the reviewer.

We believe that we had addressed all the concerns/questions raised by the reviewer and hope that the manuscript is now acceptable to Children journal.

Unfortunately, the authors' answers to my comments were not convincing, the most important part of this article was not modified / supplemented with new data, I am referring to the results chapter. The authors present, as I have said some simple statistical analyzes, i.e. SD, p. and F, I think it is very little for an article and for such a reputable journal. The authors justified that they cannot add other data because they have other articles in evaluation, the motivation is not enough, I consider that the results of a research are essential and especially the degree of complexity for a scientific article. Taking into account these aspects, I consider that the article must be reviewed again and completed with what I requested.

Our response: We fully understand your concern about the quality of the paper. We really tried to add, in this moment, the only possible analysis we could, Effect size and confidence interval. We have only two age groups in this research, so we really presented what we had. Nevertheless, we think, again, that this is valuable information having in mind that there were no other studies conducted in Serbia that could give similar information. We would really appreciate your positive comments regarding our research. On the other side, please suggest any solution for our problem, and we will be happy to implement it in our manuscript.

The authors also said that they changed the presentation of the references, if we look at the references they do not agree with the policy of the journal, see the abbreviation of the journal and others.

Our response: We have used the correct template and references. Thank you.

Also, I did not understand the explanation regarding the fact that some children arrive at kindergarten at 6.00, please a new detail, I think it is very hard to believe to get to kindergarten at that time, especially for children.

Our response: It was clearly stated that this was an option because in Serbia some people work from 7 o clock, and there were requests to start the arrival at 6 o clock. However, a great majority of children come around 7:30 o clock.

This manuscript is a resubmission of an earlier submission. The following is a list of the peer review reports and author responses from that submission.

Round 1

Reviewer 1 Report

Introduction:

The authors do a nice job of describing the layout of the study. In some areas a more in-depth dive into the content would be beneficial (i.e., lines 27-34 and 50-58).

The English is a bit difficult to read in some areas of the introduction.

Method

The authors do a nice job of clearly explaining the participants for this study. However, more information specific to the setting is needed to further analyze this paper.

What is the setting like in these daycares? What are the facilities the children have access to? What is the typical day schedule? These help to differentiate differences and are important.

Procedures: Can more information be given as to how pedometers were given out, how long they wore them for?

Results:

Data is displayed well. Explanation is good and clear.

Discussion/Conclusions:

The authors need to provide a deeper analysis of the findings. The authors do a good job discussing the previous literature, but do not provide rationale about the current findings, and why they feel they occurred. There also must be better implications for the field/future research. This section appears at first and second read fairly basic.

Author Response

December 31, 2020

Re: Children-1044353

To the Editors

Dear Editor,

Please find attached the revised version of our manuscript children-1044353, entitled “Physical Activity of Serbian Children in Daycare”

We thank the reviewers for their careful evaluation and helpful comments to our manuscript. We have carefully taken their comments into consideration in preparing our revision, which has resulted in a paper that is clearer, broader and more compelling.

Some changes were made to this paper in order to improve it. Please find below our point-by-point responses to each of the comments of the reviewers. All changes are marked in the text.

We hope that the revisions in the manuscript and our accompanying responses will be sufficient to make our manuscript suitable for publication in Children.

Sincerely,

Maja Batez

Reviewer 1

Thank you for your detailed review of our manuscript and for providing some insightful and thought-provoking suggestions to strengthen our manuscript. We feel we have sufficient responses to each of your major concerns listed above, which are further detailed below, and hope that they alleviate the concerns you have regarding the approaches adopted in our manuscript.

Introduction:

The authors do a nice job of describing the layout of the study. In some areas a more in-depth dive into the content would be beneficial (i.e., lines 27-34 and 50-58).

Our response: Thank you for comment and suggestion. The mentioned parts were revised.

The English is a bit difficult to read in some areas of the introduction.

Our response: Thank you. We have try to improve English and native speaker red and the edited manuscript.

Method

The authors do a nice job of clearly explaining the participants for this study. However, more information specific to the setting is needed to further analyze this paper.

Our response: We agree with you comments that daycare settings is important having in mind they wore pedometers during day care. We have revised the methods part accordingly.

What is the setting like in these daycares? What are the facilities the children have access to? What is the typical day schedule? These help to differentiate differences and are important.

Our response: We have added the usual settings in daycare and activities during the day. The changes can be seen in the methods part in revised manuscript.

Procedures: Can more information be given as to how pedometers were given out, how long they wore them for?

Our response: Sorry for omitting that information. The data was provided in the methods part.

Results:

Data is displayed well. Explanation is good and clear.

Our response: Thank you for your comment, we appreciate that sincerely.

Discussion/Conclusions:

The authors need to provide a deeper analysis of the findings. The authors do a good job discussing the previous literature, but do not provide rationale about the current findings, and why they feel they occurred. There also must be better implications for the field/future research. This section appears at first and second read fairly basic.

Our response: We agree with your comment and therefore included some important explanations considering our results. Please find the corrections in the discussion and conclusion part.

Reviewer 2 Report

First of all, I would like to thank editor for the opportunity to review the article entitled « Physical Activity of Serbian Children in Daycare « . The article has potential to be published but some major revisions are needed in order to be published.

Abstract :

Please, improve some expressions (i.e., « as well as the population of children »...). Improve the conclusion in the abstract

Introduction :  

  1. “and with proper nutrition, it represents the most important factor to avoid obesity that causes the emergence of many health problems » A better explanation is needed.
  2. Please, explain the following sentence : « Preschool children were once seen as extremely active... ».
  3. Authors need to define sedentary behavior
  4. Lines 44-45: Please, review the new WHO recommendations.
  5. This is very important : « Active participation of educators in the game, more frequent involvement of parents in the 50 physical activity of children, and an adequate behavioral model... » but it seems that authors have not assessed any parents behaviors.
  6. Kindergarten staff plays a significant role because a positive correlation between their education and personal engagement with children's physical activity has been established. This sentence need to be explain in detail.
  7. Lines 60-61 : Please, give more information about the sentence included.
  8. Why did authors choose pedometers to assess physical activity ? They need to justify in detail the selection chosen.

Method :

Is there any ethical approval number ?

Lines 93-95 : Why did they choose only 5 days ? Was there any failure in the data collection ?

Lines 103-105 : Please, give further information about the parametric and non-parametric characteristics of data.

I would like to know the intensity of the steps.

Is there any socioeconomic status that might be a covariable ?

Discussion

Please, introduce the aims and hypothesis to test in the study.

Is there any other activities that should be explained in the discussion ?

Please, more information about the environmental factors (lines 179-181) are needed.

Please, give more information about the increasing of physical activity in girls (lines 193-195).

Author Response

December 31, 2020

Re: Children-1044353

To the Editors

Dear Editor,

Please find attached the revised version of our manuscript children-1044353, entitled “Physical Activity of Serbian Children in Daycare”

We thank the reviewers for their careful evaluation and helpful comments to our manuscript. We have carefully taken their comments into consideration in preparing our revision, which has resulted in a paper that is clearer, broader and more compelling.

Some changes were made to this paper in order to improve it. Please find below our point-by-point responses to each of the comments of the reviewers. All changes are marked in the text.

We hope that the revisions in the manuscript and our accompanying responses will be sufficient to make our manuscript suitable for publication in Children.

Sincerely,

Maja Batez

Reviewer 2

First of all, I would like to thank editor for the opportunity to review the article entitled « Physical Activity of Serbian Children in Daycare «The article has potential to be published but some major revisions are needed in order to be published.

Our response: Thank you for your detailed review of our manuscript and for providing some insightful and thought-provoking suggestions to strengthen our manuscript. We feel we have sufficient responses to each of your major concerns listed above, which are further detailed below, and hope that they alleviate the concerns you have regarding the approaches adopted in our manuscript.

Abstract :

Please, improve some expressions (i.e., « as well as the population of children »...). Improve the conclusion in the abstract

Our response: We agree with your comment and changed the requested part accordingly. We hope the conclusion in the manuscript is now clearer and stronger.

Introduction:  

  1. “and with proper nutrition, it represents the most important factor to avoid obesity that causes the emergence of many health problems » A better explanation is needed.

Our response: The sentence was rewritten. We hope that now is cleared and easier to understand.

  1. Please, explain the following sentence: « Preschool children were once seen as extremely active... ».

Our response: Thank you for your comment. The sentence was revised. What we meant to say was that before studies conducted with valid instruments, preschool age was considered as very active age.

  1. Authors need to define sedentary behavior

Our response: We added the definition for sedentary behaviour.

  1. Lines 44-45: Please, review the new WHO recommendations.

Our response: We have included the newest WHO recommendations.

  1. This is very important: « Active participation of educators in the game, more frequent involvement of parents in the 50 physical activity of children, and an adequate behavioral model... » but it seems that authors have not assessed any parent’s behaviors.

Our response: We are sorry for omitting this. It was listed as a limitation in our study.

  1. Kindergarten staff plays a significant role because a positive correlation between their education and personal engagement with children's physical activity has been established. This sentence need to be explain in detail.

Our response: The sentence was revised, thank you.

  1. Lines 60-61: Please, give more information about the sentence included.

Our response: Sorry for not using the proper explanation. The sentence was rewritten and can be seen in our revised manuscript.

  1. Why did authors choose pedometers to assess physical activity? They need to justify in detail the selection chosen.

Our response: There is a strong evidence that daily step counts in preschool children give valid information on physical activity levels. This was determined by comparing the pedometers data and results from accelerometers (Cardon, & De Bourdeaudhuij, 2007) and confirmed later (Pagels et al, 2011).

Cardon, G., & De Bourdeaudhuij, I. (2007). Comparison of pedometer and accelerometer measures of physical activity in preschool children. Pediatric exercise science, 19(2), 205-214.

Pagels, P., Boldemann, C., & Raustorp, A. (2011). Comparison of pedometer and accelerometer measures of physical activity during preschool time on 3‐to 5‐year‐old children. Acta paediatrica, 100(1), 116-120.

Method:

Is there any ethical approval number?

Our response: The ethical approval number was added according to Editors comments. Therefore, maybe it was added by editors in newer version of the manuscript. Nevertheless, now it stands in the methods part.

Lines 93-95: Why did they choose only 5 days Was there any failure in the data collection?

Our response: Thank you for your question. Number of days was 5 because the kindergarden only works in those days. Having in mind that we only measured the kindergarden activities it was logical to choose this days only.

Lines 103-105: Please, give further information about the parametric and non-parametric characteristics of data.

Our response: We are sorry for omitting that information. It was added in the Statistical analysis part.

I would like to know the intensity of the steps.

Our response: We are sorry, but this information was used for another paper that is in the publication process so we did not included the intensity of steps in this paper. Thank you for noticing this important fact.

Is there any socioeconomic status that might be a covariable?

Our response: This was added as a limitation for this study.

Discussion

Please, introduce the aims and hypothesis to test in the study.

Our response: We have added the hypothesis in the introduction and discussion and clearly defined aims.

Is there any other activities that should be explained in the discussion?

Our response: We have explained the daycare settings and activities in the methods section. Additionally, we have explained some reasons for our results regarding activities in daycare and the advantages of outdoor activities.

Please, more information about the environmental factors (lines 179-181) are needed.

Our response: As mentioned in previous comment, environmental factors and their impact on results were explained additionally. We think and the results have confirmed that this factors are of great importance during daycare.

Please, give more information about the increasing of physical activity in girls (lines 193-195).

Our response: Thank you for noticing that. We agree and added some explanation and possible reasons for differences between boys and girls.

Round 2

Reviewer 1 Report

Thank you for making some of the changes. There are still some minor grammatical issues and overall English changes that must be made. 

Author Response

Thank you for giving us the opportunity to submit a revised draft of our manuscript titled “Physical Activity of Serbian Children in Daycare”. We  appreciate the time and effort that you and other reviewers have dedicated to providing your valuable feedback on our manuscript. We are grateful to the reviewers for their insightful comments on our paper. We have been able to incorporate changes regarding English grammatical errors and overall English. The manuscript was English edited by native speaker.

Reviewer 2 Report

Authors have improved the quality of the manucript, but in my opinion there are some major aspects that should be modified in order to accept the article.

Abstract

Did authors measure sex or gender ? Please, clarify this issue.

Introduction

What type of physical exercise/activity is important for children ? (moderate, vigorous ?) (How long ?) (Lines 28-31)

Is it important to assess children´s physical activity levels or their parents´ physical activity levels ?

It is important to note that in early ages is difficult to conduct physical activity by theirselves.

Instrument

Please, give further information or support the idea that pedometer can measure physical activity. What type of physical activity ? Vigorous ?

The problem is that authors give information about a typical day, but they did not indicate what children do in that time.

Statistical Analysis

Please, could authors provide K-S results or other techniques used in order to know the use of parametric techniques ?

Results

The problem is that authors did not control the activitis themselves, so justify the differences is quite hard. Please, give some information about the activities, extra-curricular activities...

Please, explain Table 3.

Discussion

The discussion must start with the aim of the study in order to give that information to readers.

Please, give further information about kindergarden curriculum.

Maybe, the low number of participants could make influence on the results.

Lines 229-236 are nos based on scientific information.

If authors have not controlled by parents, for example, it is quite difficult to discuss the data, because at the age of 4 years old, for example, physical activity levels depend on parents´ physical activity (or decisions).

Author Response

Thank you for providing us this opportunity to further revise our manuscript entitled Physical Activity of Serbian Children in Daycare.

We appreciate the very positive and constructive comments from the reviewer.

We believe that we had addressed all the concerns/questions raised by the reviewer and hope that the manuscript is now acceptable to Children journal. Should you have any additional requests or questions, please do not hesitate to contact me.

Reviewer

Authors have improved the quality of the manuscript, but in my opinion there are some major aspects that should be modified in order to accept the article.

Our response: We thank the reviewer’s very positive comment. According to the reviewer’s suggestion, we have modified and improve the requested parts of the manuscript. We hope that revised manuscript is appropriate for publishing.

Abstract

Did authors measure sex or gender ? Please, clarify this issue.

Our response: We used gender, which is more appropriate for current variables and according to similar studies also.

Trost, S. G., Pate, R. R., Sallis, J. F., Freedson, P. S., Taylor, W. C., Dowda, M., & Sirard, J. (2002). Age and gender differences in objectively measured physical activity in youth. Medicine and science in sports and exercise34(2), 350-355.

Louie, L., & Chan, L. (2003). The use of pedometry to evaluate the physical activity levels among preschool children in Hong Kong. Early Child Development and Care173(1), 97-107.

Jackson, D. M., Reilly, J. J., Kelly, L. A., Montgomery, C., Grant, S., & Paton, J. Y. (2003). Objectively measured physical activity in a representative sample of 3‐to 4‐year‐old children. Obesity research11(3), 420-425.

Introduction

What type of physical exercise/activity is important for children ? (moderate, vigorous ?) (How long ?) (Lines 28-31)

Our response: Thank you. It was corrected.

Is it important to assess children´s physical activity levels or their parents´ physical activity levels ?

Our response: Of course it is more important to assess the children’s PA levels. However, in preschool age, the activity of parents is of great importance. Therefore, we have only stated this in the introduction without further deeper analysis.

It is important to note that in early ages is difficult to conduct physical activity by theirselves.

Our response: We agree with your comment about organized physical activity. Therefore, we stated that the educators have a big responsibilities in this age. However, during free play and activities, children are often

Instrument

Please, give further information or support the idea that pedometer can measure physical activity. What type of physical activity ? Vigorous ?

Our response: We are sorry for omitting that information. We have provided the requested information in the procedure part.

The problem is that authors give information about a typical day, but they did not indicate what children do in that time.

Our response: We have added information about a typical day that could give a better understanding of activities in daycare.

Statistical Analysis

Please, could authors provide K-S results or other techniques used in order to know the use of parametric techniques ?

Our response: The use of K-S test and the information about normality of data are provided in statistical analysis. If the reviewer request, we can provide the results for every variable in tables. However, due to earlier studies experience, the information about normality of data is enough.  

Results

The problem is that authors did not control the activitis themselves, so justify the differences is quite hard. Please, give some information about the activities, extra-curricular activities...

Our response: We understand the idea of the reviewer to find the significant predictor that influence the difference in the number of steps during daycare. Unfortunately, we didn’t examined the predictors and activities during daycare separately. Our idea was just to investigate if there are differences in number of steps during daycare in children regarding age, gender and the frequency of going out. However, we are thankful for that suggestion and comment and we will surely take this into account for future studies.

Please, explain Table 3.

Our response: Explained additionally.

Discussion

The discussion must start with the aim of the study in order to give that information to readers.

Our response: We are sorry for that mistake. We have added the aim and the main outcomes.

Please, give further information about kindergarden curriculum.

Our response: thank you for your suggestion. We have added information about curriculum, kindergardens and the importance of conditions in kindergardens for physical activity. The conditions in the current study were very similar and thanks to that we could conclude regardless of mentioned conditions.

Maybe, the low number of participants could make influence on the results.

Our response: We agree with your comment. We have conducted g power analysis to detect the minimal sample for our study. According to analysis, we have minimal number of participants to detect possible differences on the level of 0.05. Nevertheless, we agree that bigger sample would give more significant insight into differences.

Lines 229-236 are nos based on scientific information.

Our response: It was corrected

If authors have not controlled by parents, for example, it is quite difficult to discuss the data, because at the age of 4 years old, for example, physical activity levels depend on parents´ physical activity (or decisions).

Our response: The influence of parents is the biggest in this age group, of course. Therefore, we have conducted the measurement only during the time where children are in daycare. Moreover, we have excluded the weekends. So, the parents have greatest influence on overall children’s physical activity. However, during daycare, children are isolated with other children. The only responsibility goes to children and partially to the educators who can significantly improve the physical activity of children, but also the motivation to be physically active in leisure time and throughout the life.